# Green Synthesis and Characterization of Pullulan Mediated Silver Nanoparticles through Ultraviolet Irradiation

**DOI:** 10.3390/ma12152382

**Published:** 2019-07-26

**Authors:** Muhammad Jamshed Khan, Suriya Kumari, Kamyar Shameli, Jinap Selamat, Awis Qurni Sazili

**Affiliations:** 1Institute of Tropical Agriculture and Food Security, Universiti Putra Malaysia (UPM), Serdang 43400, Malaysia; 2Faculty of Veterinary Sciences, Bahauddin Zakariya University, Multan 60800, Pakistan; 3Malaysia-Japan International Institute of Technology, Universiti Teknologi Malaysia, Jalan Sultan Yahya Petra, Kuala Lumpur 54100, Malaysia

**Keywords:** pullulan, silver nanoparticles, UV irradiation, green synthesis

## Abstract

Nanoparticles (NPs) are, frequently, being utilized in multi-dimensional enterprises. Silver nanoparticles (AgNPs) have attracted researchers in the last decade due to their exceptional efficacy at very low volume and stability at higher temperatures. Due to certain limitations of the chemical method of synthesis, AgNPs can be obtained by physical methods including sun rays, microwaves and ultraviolet (UV) radiation. In the current study, the synthesis of pullulan mediated silver nanoparticles (P-AgNPs) was achieved through ultraviolet (UV) irradiation, with a wavelength of 365 nm, for 96 h. P-AgNPs were formed after 24 h of UV-irradiation time and expressed spectra maxima as 415 nm, after 96 h, in UV-vis spectroscopy. The crystallographic structure was “face centered cubic (fcc)” as confirmed by powder X-ray diffraction (PXRD). Furthermore, high resolution transmission electron microscopy (HRTEM) proved that P-AgNPs were covered with a thin layer of pullulan, with a mean crystalline size of 6.02 ± 2.37. The average lattice fringe spacing of nanoparticles was confirmed as 0.235 nm with quasi-spherical characteristics, by selected area electron diffraction (SAED) analysis. These green synthesized P-AgNPs can be utilized efficiently, as an active food and meat preservative, when incorporated into the edible films.

## 1. Introduction

Metal nanoparticles (NPs) have been frequently investigated by the researchers for their multi-dimensional applications after their synthesis in 1959 by Richard Feynman, with a size ranging from 1 nm to 100 nm, [1,2,3]. NPs exhibited a new behavior of efficiency at even very low concentration [4]. The recent trend of nanotechnology research has been diverted to antimicrobial food packaging, delivery of nano-medicines/drugs, gene delivery vectors, nano-imaging and Biosensors for cancer diagnosis, and polymeric nano-composite wound dressing [3,4].

Among metal NPs, the utilization of silver nanoparticles (AgNPs) in the biomedical sector, animal and human nutrition, electronic industry and food preservation is remarkable [5]. AgNPs, as compared to other metal NPs, exhibit very low volatility, with tolerance against high temperatures as compared to other nanoparticles [6]. Moreover, AgNPs can restrain the growth of the microorganisms (including Gram positive and Gram negative bacterial strains; fungi) from the initial contact and further penetration into the cellular membrane of the microbes [7,8].

Various chemical and physical techniques have been reported for the synthesis of AgNPs. The chemical techniques have certain limitations including strict reaction protocols, toxic reagents, time consumption and unstable NPs [9]. A new and relatively safer concept has been presented by Raveendran et al. [10] as the “green synthesis” of AgNPs. The synthesis of AgNPs, with reduced size and better shape, through direct physical techniques, is another green and promising sector [11]. These physical techniques mainly include synthesis by sun rays, microwave radiation, gamma radiation or ultraviolet (UV) irradiations [9,12,13]. For the physical methods, the toxicity, economy and benign biological nature of the compounds should not be ignored [14,15,16]. The organic biopolymers are economical, environment friendly and exhibit no or very low toxic nature for the physical synthesis of AgNPs [12,13,17,18,19,20,21]. These organic biopolymers mainly include the exo-polysaccharides (EPS) obtained from bacteria, algae and fungi [10,22,23,24].

Pullulan is an extra cellular polysaccharide produced by a fungal species *Aureobasidium pullulans* [25]. For the synthesis of AgNPs, few studies have been reported to utilize pullulan as a green reducing, capping and stabilizing agent [26,27,28,29]. Pullulan mediated silver nanoparticles (P-AgNPs) can be applicable in food and meat preservation, drug carrying and therapeutic studies [30]. Ganduri et al. [31] reported that AgNPs synthesized from the increasing concentrations of pullulan, improved the formation of AgNPs with better quality and quantity [11,31].

The direct and green synthesis of AgNPs mediated by pullulan (reducing and capping agent), through UV irradiation, has not been reported yet. Thus, the present study was planned to use UV irradiations, as a physical green technique, for the green synthesis of P-AgNPs. The impact of increasing time of UV irradiation (from 0 h to 96 h) on the synthesis, size and distribution of AgNPs was also studied.

## 2. Materials and Methods

### 2.1. Materials

The materials for the synthesis of P-AgNPs were pullulan (97.0% pure, food grade, obtained from *Aureobasidium pullulans*; Sigma-Aldrich, St. Louis, MO, USA), Silver nitrate (AgNO_3_) (R & M, London, UK) and deionized water (Sigma-Aldrich). All the reagents used in the study were of analytical grade and no purification was conducted before their use. Pullulan was dried at 100 °C for 24 h before use.

### 2.2. Green Synthesis of Pullulan Mediated Silver Nanoparticles

Three concentrations of pullulan were prepared as per the procedure reported by Shameli et al. [13] with modifications. The 1.0, 2.0 and 4.0 g of pullulan powder, equal to 5, 10 and 20% (w/v),were stirred with 200 mL of deionized water for 1.0 h at 500 rpm at 24–25 °C.

A clear solution was obtained after 60 min of stirring. Three concentrations of AgNO_3_ aqueous solution (0.02, 0.50, and 1.0 M) were produced by dissolving the calculated masses of AgNO_3_ in deionized water. To synthesize P-AgNPs, 250 mL of each pullulan solution (5, 10, and 20 wt.%) was mixed with an equal volume of the aqueous solutions of AgNO_3_. The assorted solutions were stirred thoroughly for 1.0 h at 65 °C for 500 rpm to complete the chemical reaction.

After stirring, the obtained suspensions were clear but viscous. These suspensions were put under ultraviolet radiation with a wavelength of 365 nm and the magnetic stirring speed of 195 rpm. UV irradiation was done by an ultraviolet reactor (ChECA, UTM, Kuala Lumpure, Malaysia) with a power of 450 W. The feature of the UV-reactor bulb was UV-C Radiation (200 to 280 nm) (Philips, TUV 8W, G8 T5, Kuala Lumpure, Malaysia). The irradiation process was done for 1, 3, 6, 8, 12, 24, 48, 64, and 96 h (A1–A9). After UV irradiation, the suspensions were centrifuged 25 °C for 30 min to obtain the pellets of P-AgNPs.

### 2.3. Characterization of Pullulan Mediated Silver Nanoparticles

Pullulan mediated silver nanoparticles (P-AgNPs) were characterized [9,12,13] through ultraviolet-visible spectroscopy (UV-vis) by a U-5100, UV-vis spectrophotometer (Hitachi, High Technologies Co., Tokyo, Japan) with a wavelength range of 220 nm to 800 nm. The analysis of the structure was conducted through powder X-ray diffraction (PXRD) (PANanalytical EMPYREAN, Eindhoven, The Netherlands), Xpert High score Plus (V.14.0; PANanalytical EMPYREAN) at a lower angle of 30°–80° and scan with a speed of 2°/min. The topographic analysis was performed by field emission scanning electron microscopy (FESEM) using a JSM 7600 F FESEM (JEOL Ltd., Tokyo, Japan). High-resolution transmission electron microscopy (HRTEM) was conducted through a JEM 2100F filed emission electron microscope (JEOL Ltd.) with a high resolution of 200 kV using a LaB_6_ electron gun. Zeta potential of P-AgNPs was confirmed by a Litesizer 500, particle analyzer (Anton paar USA Inc., Ashland, VA, USA) while Fourier transform infrared spectroscopy (FT-IR) was conducted, with the spectral range of 500 cm^−1^ to 4000 cm^−1^, by using an IRTracer 100 Fourier transform infrared spectrophotometer (SHIMADZU Corp., Kyoto, Japan). The colloidal solution was centrifuged through a refrigerated micro-centrifuge 5810 R (Sigma-Aldrich) to obtained P-AgNPs pellets.

Most of the characterization was performed with the colloidal solution dissolved in distilled water, while P-AgNPs powder was used for PXRD and FESEM analyses. Statistical analysis of the data was done by using SPSS software to measure the mean crystalline size (V.20.0, IBM Corp., Armonk, NY, USA).

## 3. Results and Discussion

In the present study, AgNPs were synthesized by UV irradiation successfully and pullulan was used as “stabilizer”. Further, the impact of irradiation time on the formation of NPs along with the size of NPs was also considered. Pullulan 10% (w/v) solution along with 0.50 M AgNO_3_ aqueous solution showed the formation of P-AgNPs. The formation of P-AgNPs was confirmed by the dark color of the colloidal suspension, after 96 h of UV irradiation [13] (Figure 1).

The formation of NPs was due to the reduction of AgNO_3_, changing the color of clear colloidal solution into light gray, then gray, and finally darker gray [13,32]. Our results are in agreement with the results as reported by Kanmani and Lim [32], Shameli et al. [13], and Saravanan et al. [24] for the formation of AgNPs by the reduction of AgNO_3_. The chemical structure of pullulan consists of repeating units of malto-triose connected with ά*-(1-6)* linkages, providing solubility and flexibility to pullulan [33]. During the reduction process, the produced AgNPs were attracted by the amino radicals (NH_2_) and hydroxyl (OH) ends of malto-triose units of pullulan [13]. As a result, pullulan acts as a strong protective layer and a highly functional coating for AgNPs, with better stability in the solution over the time [30].

The impact of pullulan, as reducing and capping agent, on the formation of P-AgNPs was proven by ultraviolet visible (UV-vis) spectroscopy after various irradiation times (Figure 2). The UV-vis spectroscopic peaks were recorded from wavelengths 300–800 nm, where NPs formed a broad surface plasma resonance ultraviolet absorption band in the wavelength range of 380–480 nm [24].

It was also observed that P-AgNPs, formed in the colloidal solution after 96 h of irradiation, expressed a wavelength peak at 415 nm. It was due to the fact that the AgNPs absorb UV radiation in the visible range of electromagnetic spectrum of 380 nm to 450 nm, because of surface plasmon resonance (SPR) transition of AgNPs [17,34].

Therefore, based on the achieved results, pullulan with 10% concentration showed the formation of P-AgNPs, confirmed by the UV-vis spectra, indicating the impacts of pullulan concentrations, as well as UV irradiation time, on AgNPs formation (Figure 2). Pullulan mediated silver nanoparticles (P-AgNPs) formed after 24 h from UV-irradiation time, and the peak intensities were observed after 48, 64 and 96 h (Figure 2). This long-term study was related to evaluate the effect of radiation on the size and distribution of AgNPs. During that period we witnessed two phenomena at different times: contained aggregation at the beginning, and disaggregation at the end of radiation time. It was confirmed that the size of NPs was also reduced as the irradiation time increased from 0 h to 96 h. The power consumption (power × time), during irradiation, was recorded as 10.80 (at 24 h) to 43.20 (at 96 h) kWh/500 mL of colloidal solution for P-AgNPs synthesis (Table 1, Figure 2C).

The UV irradiation induced reduction process has been provided below:
(1)$${\mathrm{AgNO}}_{3}\overset{\text{UV-irradiation}}{\to }\left({\mathrm{Ag}}^{+}\right)+{{\mathrm{NO}}_{3}}^{-}$$
(2)$$\left({\mathrm{Ag}}^{+}\right)+e¯\left(\mathrm{eq}\right)\overset{\mathrm{reduction}}{\to }\left(\mathrm{Ag}\right){}^{\circ}$$
(3)$$\left(\mathrm{Ag}\right){}^{\circ}+\left({\mathrm{Ag}}^{+}\right)\to {\left({\mathrm{Ag}}^{+}\right)}^{2}$$
(4)$$n\left({\mathrm{Ag}}^{+}\right)+{\left({\mathrm{Ag}}^{+}\right)}^{2}\to {\left({\mathrm{Ag}}^{+}\right)}^{n}$$
(5)$${\left({\mathrm{Ag}}^{+}\right)}^{n}+e¯{\left(\mathrm{eq}\right)}^{n}\to {\left(\mathrm{Ag}\right)}^{n}$$

$\left({\mathrm{Ag}}^{+}\right)$ is the nano-cluster of the silver obtained from silver precursor (AgNO_3_) and e¯(eq) are the electrons in the aqueous solution. After UV irradiation of pullulan/AgNO_3_ colloidal solution, Ag^+^ was, first, reduced to (Ag)°, then ${\left({\mathrm{Ag}}^{+}\right)}^{n}$, leading towards the formation of ${\left(\mathrm{Ag}\right)}^{n}$ by the large number of aqueous electrons produced during the process [13]. Ultraviolet light covers a wave length spectrum from 100 to 380 nm and is sub-divided into three regions by wavelength including UV-A (320–400 nm), UV-B (280–320 nm), and UV-C (200–280 nm). UV-C is considered as the most energetic ultraviolet radiation [11]. Therefore, in this study, UV-C was used for the reduction of Ag^+^ to ${\left(\mathrm{Ag}\right)}^{n}$.

It was possible to control the synthesis, size, and the shape of AgNPs surrounded by pullulan, by irradiation time as expressed by UV-vis spectra [12]. UV irradiation induced the reduction of the AgNPs precursor in the presence of pullulan solution [12,13]. Photo-chemical induced reduction of AgNO_3_ into AgNPs, in the presence of a polysaccharide capping agent, can not only control the sizes of NPs, but also their shapes [35].The prolonged exposure of AgNO_3_ colloidal solution containing silver cation ions causes the heating of AgNPs, resulting in their breakdown into smaller sized NPs [35]. As the irradiation time increased, the size of the synthesized AgNPs was reduced. This can be considered as the irradiation time dependent factor to control the synthesis, size, and shape of nanoparticles [36,37].

Our results are in agreement with Pandey et al. [34], Esumi et al. [35], Kuthirummal et al. [36], and Huang et al. [37] with respect to irradiation time controlling the size and shape of NPs.

The crystalline nature along with the topography of P-AgNPs, synthesized after 96 h of irradiation, was further confirmed by powder X-ray diffraction (PXRD). It can be observed that the crystalline nature of P-AgNPs was confirmed as crystalline silver (Figure 3). The nanoparticles exhibited four silver peaks in PXRD analysis at 2 theta/degree, i.e., 111 (38.17°), 200 (44.33°), 220 (64.49°), and 311 (77.44°). These peaks are consistent with Ag° Peak Ref. No. 00-004-0783 (99.9% pure silver; fcc) confirming their topographic nature as “face centered cubic (fcc)” [30]. From PXRD results, it is possible to estimate the average crystalline size of P-AgNPs by using the Debye-Scherrer’s equation which is given as shown below;
(6)n = Kλ/βcos θ
where ***n*** is the mean crystalline size, *K* is Sherrer’s constant (shape factor) with value 0.94 for spherical crystallites with cubical symmetry, *λ* is X-ray wavelength (1.54 Å; for powder samples only), *θ* is the Bragg angle and *β* is the line broadening at half of the XRD peak [38]. The confirmation of actual crystalline size was conducted by high resolution transmission electron microscopy (HRTEM). Nevertheless, PXRD patterns are the main characteristic of P-AgNPs, also in concordance with the findings of Coseri et al. [30]; Bankura et al. [17]; Kanmani and Lim, [32] and Roopan [20].

Similar findings have been reported for the typical PXRD spectra of AgNPs with four diffraction peaks (111, 200, 220, 311) by Coseri et al. [30]. A four diffraction peak pattern of AgNPs confirms the “fcc” topographic nature of NPs [17,30,32]. Bankura et al. [17] and Roopan [20] had previously reported that the four peak pattern at 2-theta/degree could be due to the crystallization of reducing and capping agent used for the synthesis of Ag NPs.

The topographic structure and shape of P-AgNPs was confirmed as “spherical” by field emission scanning electron microscopy (FESEM) (Figure 4A,B). The FESEM micrographs of P-AgNPs revealed the presence of spherical shaped P-AgNPs at 10,000×. Our study was in agreement with the findings of Saravanan et al. [24], for the distribution of spherical shaped “bacterial exo-polysaccharide AgNPs. The mean crystalline size of NPs was observed as 42.27 ± 12.2 nm after 48 h of UV irradiation (Figure 4C) and 6.02 ± 2.37 nm after 96 h of UV irradiation (Figure 4D), as confirmed by high resolution transmission electron microscopy (HRTEM). The decrease in crystalline size of P-AgNPs from 48 h to 96 h of UV irradiation, was proven by HRTEM analysis, along with “quasi-spherical” characteristics, as per reports by Quester et al. [39]. The typical “quasi-spherical” nature is one of the prominent shape characteristics of AgNPs for their application in various fields [40]. By HRTEM observation at 5 nm, it was found that P-AgNPs were delimited by a thin layer of an organic material (Figure 4E) which is supposed to be the reducing and the capping agent pullulan [40]. The top to top distance, between the surface lattice of AgNPs, was 0.235 nm (Figure 4E) in selected area electron diffraction (SAED) analysis (Figure 4F) [41].

The possible involvement of the functional groups of pullulan for the synthesis of P-AgNPs was studied by FT-IR spectra. Pullulan aqueous solution (10% w/v, without UV irradiation) was compared with pullulan + AgNO_3_ colloidal solution (with 96 h of UV irradiation, A9). It can be observed that P-AgNPs expressed their characteristic peaks from wave numbers 3925 cm^−1^ to 1325 cm^−1^ (Figure 5a). The first strong and broader peak was at 3359 cm^−1^ (Figure 5a) indicating the stretching and involvement of hydroxyl (O–H) group of pullulan [31].

The O–H group of pullulan is considered as an efficient coordinator with silver ions of P-AgNPs during the reduction process [34]. The prominent peaks were at wave numbers 2418, 2167, 1950, 1637, and 1535 cm^−1^ (Figure 5a), proving the stretching frequencies of carbonyl (C=O) and C=C groups of pullulan [31,32,42]. For the involvement and the stretching frequencies of C=O and C=C functional groups, the standard wave numbers range from 1000 cm^−1^ to 1900 cm^−1^ [31]. The FT-IR spectral results were in agreement with the reports of Ganduri et al. [31], Kanmani and Lim [32] and Kumar et al. [41] for the involvement of O–H, C=O and C=C groups of pullulan during the synthesis of P-AgNPs.

The zeta potential of P-AgNPs was confirmed as 3.27^+^ (Figure 5b), proving “stability” of the NPs in the colloidal solution at 25 °C and pH 6.34 [34]. Pullulan has an “uncharged nature” in the colloidal solution when used as the reducing and capping agent [30]. Due to the positive charge at the core, P-AgNPs exhibited positive zeta potential in the solution, with better stability [30,34]. Pullulan concentration, in the formation of P-AgNPs, can also govern zeta potential (ZP) with positive or negative charges at room temperature [30].

## 4. Conclusions

An environment friendly, economical, safer and green physical technique has been introduced for the synthesis of P-AgNPs. Pullulan was used as a “green” capping, stabilizing and reducing agent without any addition of toxic or harmful chemicals. The effect of UV irradiation time was also recorded as the main factor for controlling the synthesis, size, shape and distribution of the nanoparticles. Green synthesized P-AgNPs contained “quasi-spherical characteristics” showing 415 nm as spectra maxima with mean crystalline size of 6.02 ± 2.37 nm after 96 h of UV irradiation. The crystallographic nature of P-AgNPs was recorded as “face centered cubic” surrounded by a thin layer of pullulan. The involvement of O–H, C=O, and C=C functional groups of pullulan, during the green synthesis, was confirmed by FT-IR spectra. Zeta potential of 3.27^+^ confirmed the stability of P-AgNPs in the colloidal solution after 96 h of UV-irradiations. It is hypothesized that the green synthesized P-AgNPs can be utilized individually or incorporated in the active packaging with an antimicrobial potential due to their purity, smaller size and spherical shape.

## Abbreviations List

NPsNanoparticlesUVUltravioletP-AgNPsPullulan mediated silver nanoparticlesFESEMField emission scanning electron microscopyHRTEMHigh resolution transmission electron microscopySAEDSelected area electron diffractionPXRDPowder X-ray diffractionFT-IRFourier transform infrared spectroscopyfccFace centered cubicRpmRevolution per minutew/vWeight/volumeM solutionMolar solution

## Figures and Tables

**Figure 1 materials-12-02382-f001:**
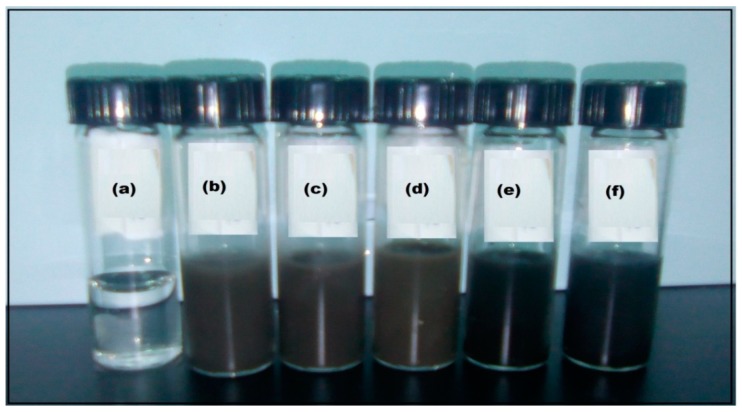
Characteristic changes in the color of pullulan solutions before and after irradiation: Pullulan/Ag ion, after 0, 12, 24, 48, 64, 96 h (a–f, respectively).

**Figure 2 materials-12-02382-f002:**
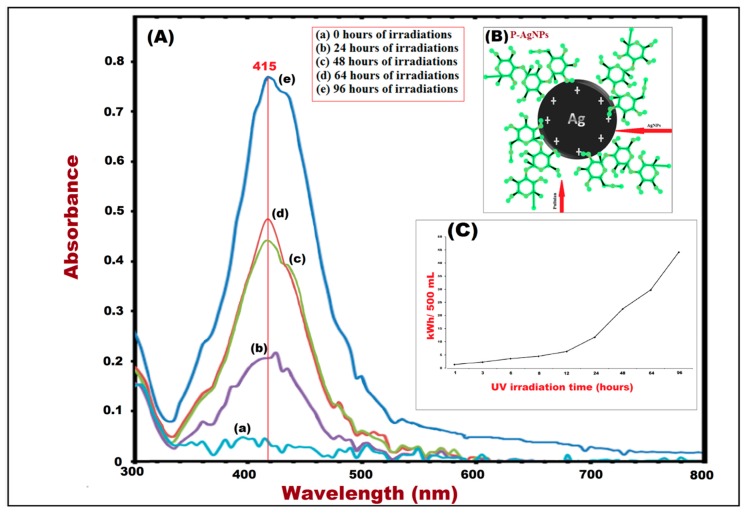
(**A**) UV-vis spectra maxima of P-AgNPs after 96 h of UV irradiation. (**B**) AgNPs surrounded by pullulan. (**C**) Power consumption curve for UV irradiation time.

**Figure 3 materials-12-02382-f003:**
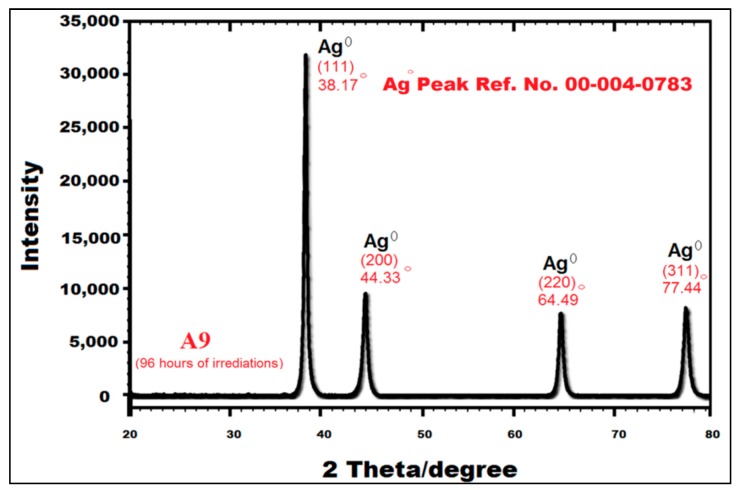
Powder X-ray diffraction (PXRD) spectra for P-AgNPs obtained after 96 h of UV irradiation.

**Figure 4 materials-12-02382-f004:**
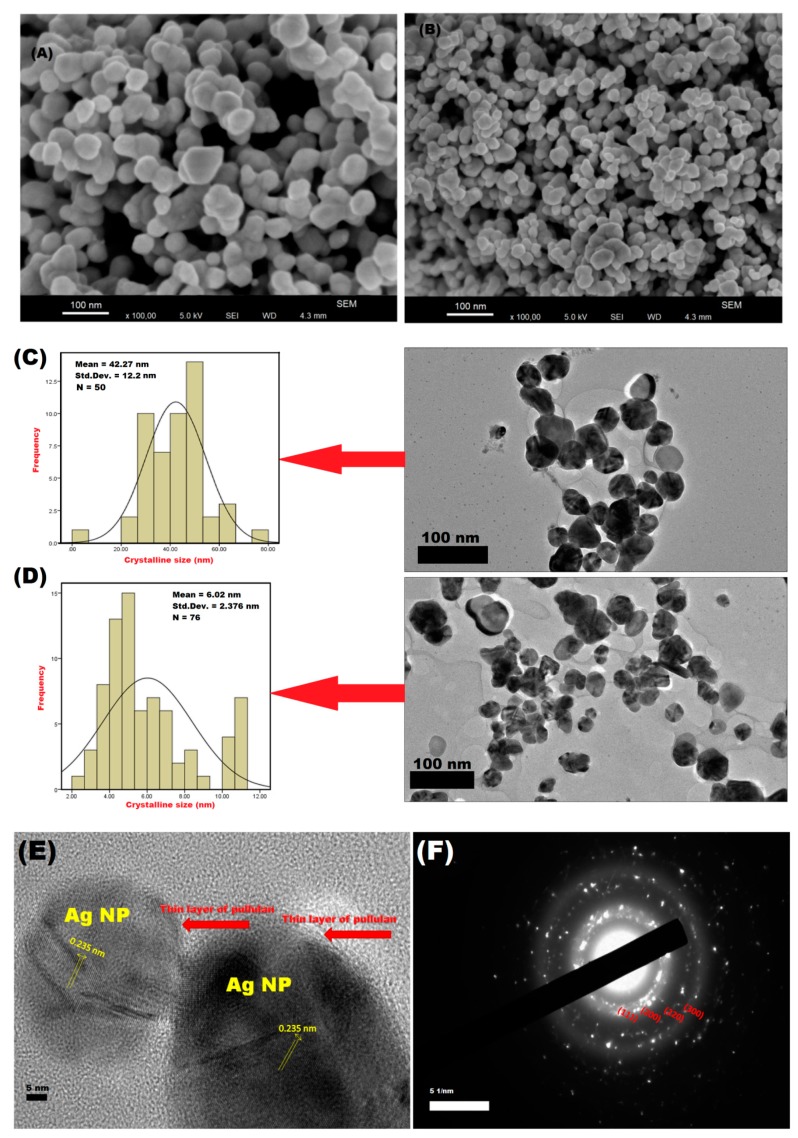
(**A**) FESEM micrograph of P-AgNPs after 48 h of UV irradiation. (**B**) FESEM micrograph of P-AgNPs after 96 h of UV irradiation. (**C**) HRTEM image showing the topography and mean crystalline size of P-AgNPs after 48 h of UV irradiation. (**D**) HRTEM image of P-AgNPs after 96 h of UV irradiation, showing the topography and mean crystalline size. (**E**) Lattice surface analysis of P-AgNPs after 96 h of UV irradiation. (**F**) Selected area electron diffraction (SAED) of P-AgNPs after 96 h of UV irradiation.

**Figure 5 materials-12-02382-f005:**
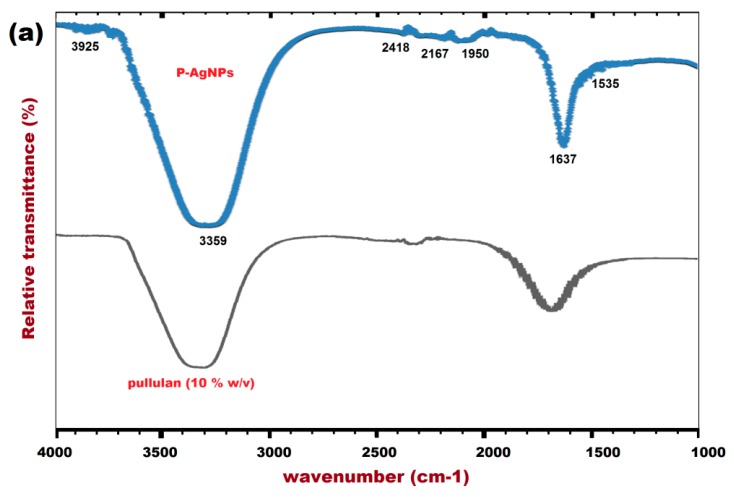
(**a**) FT-IR spectra of pullulan (10%) and P-AgNPs (**b**) Zeta potential of P-AgNPs.

**Table 1 materials-12-02382-t001:** Power consumption data for UV irradiation during the synthesis of P-AgNPs.

Samples	UV Irradiation Time (h)	Kilo Watt Hour (kWh)/500 mL
A1	1	0.45
A2	3	1.35
A3	6	2.70
A4	8	3.60
A5	12	5.40
A6	24	10.80
A7	48	21.60
A8	64	28.80
A9	96	43.20

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
