# Peer review of "Green Synthesis and Characterization of Pullulan Mediated Silver Nanoparticles through Ultraviolet Irradiation"

_materials, 2019, doi:10.3390/ma12152382_

Round 1
Reviewer 1 Report
General comments
The article deals with the experimental synthesis of silver nanoparticles (AgNPs) mediated by the natural polysaccharide polymer Pullulan.
The experiments were well designed, with sensitivity to changing working parameters rather comprehensive, the results are quite clear, the presentation and language are acceptable, despite few errors. Overall, the interest to the readers could be great enough.
However, the claim for "green synthesis" is far from confirmed, based on the available data, and on missing data.
The first problem is that the P-AgNPs (AgNPs capped by Pullulan) are formed only after 96 hours of UV irradiation, which is a long time, likely involving a great energy consumption. No power consumption data are reported, thus it is not possible to assess the specific energy consumption (energy per unit mass of... silver nitrate? silver nitrate + pullulan?).
Actually, an LCA study would be more appropriate (energy and resources consumed for production of pullulan and AgNO3, plus for production of P-AgNPs, but at this stage at least the power (energy) data for the synthesis process are needed for the sake of comparison with other processes.
The second problem lies in the laboratory-scale experiments: UV irradiation is known to have a limited penetration depth, thus what would be the requirements for the effective scalability to real-scale production processes? I fear that the requirements for UV equipment and related energy consumption would be quite high.
Answering the above-mentioned questions is mandatory for further consideration of the manuscript. As well, changes performed in response tho these questions could lead to the need of changing the Abstract and the Conclusions, too.
Specific comments
Line 4 (Title) - remove abbreviation "UV".
Line 26 - "the structure". Change to "showed that the structure".
Lines 52-55 - The Authors should clearly explain why biopolymers are needed, or useful, for the synthesis of AgNPs.
Line 57 - "produces". Change to "produced".
Line 58 - "has been reported quite limited as a green reducing and capping agent". Meaning unclear. Please rephrase.
Line 64 - "is not being". Change to "has not been".
Line 74 - "to before use." Change to "before use."
Line 75 (subsection 2.2). Check the numbers, especially percentages.
Line 77 - "5, 10 and 20 wt. %,". Percentage of what? Maybe figures should be 0.5, 1 and 2% ???
Lines 86-87 - "Any information about the consumed power?". Any information about the consumed power? (see also the First problem in General comments).
Line 122 - "units were". Remove.
Line 135 - Define SPR.
Line 137 - "based on the achieve results". Change to "based on the achieved results".
Line 144 (and elsewhere in the text) - "as under". Change to "as shown below".
Line 158 - "shape". Remove.
Lines 190-203 - Check the reference to Figures. For example, Figure 5(b) was never referred to in the text.
Lines 210-211 - "The standard solution ... for 96 hours (A9)". Unclear meaning, please rephrase.
Line 236. "maxima in after". Remove "in".
Author Response
Respectable Reviewer 1,
A very good afternoon.
Sir,
First of all, on the behalf of all the authors, I am very thankful for your precious comments regarding the manuscript. Further, I would like to inform your good self that all the suggested changes have been incorporated in the manuscript as per your kind and valuable suggestions.
The details are provided in the given table as an attachment for further Processing please.
With thanks and regards,
Jamshed.

Reviewer 2 Report
In this study authors presented a interesting application concerning nanoparticles synthesis using a green approach of microbial stabilizer as pullulan with potential in active food packaging.
I have only minor comments prior to publication.
Point 1: An abbreviation list should be included in the manuscript
Point 2: Is their any possible Reference for the procedure followed in section 2.3
Point 3: line 211-212 when you say ''was reported'' where do you refer that are actually reported?
Point 4: Figure 2 is not a common view on the Results section of a research article, usually are included figures of the work itself as you did below of F2.
Point 5: Check the whole draft for missing spaces (lines 32, 52, 59, 133, 141, 145, 149, 158, 160, 174, 176, 179, 182, 214, 215, 216, 220, 226)
Author Response
Respectable Reviewer 2,
A very good afternoon.
Sir,
On the behalf of all the authors, I am very much grateful for your precious comments regarding the manuscript. I would like to inform your good self that all the suggested changes have been incorporated in the manuscript as per your kind and valuable suggestions.
The details are provided in the given table as an attachment for further Processing please.
Thanking in anticipation,
Jamshed.
